# Two Novel Schiff Base Manganese Complexes as Bifunctional Electrocatalysts for CO_2_ Reduction and Water Oxidation

**DOI:** 10.3390/molecules28031074

**Published:** 2023-01-20

**Authors:** Xin Zhao, Jingjing Li, Hengxin Jian, Mengyu Lu, Mei Wang

**Affiliations:** 1School of Materials Science and Engineering, Institute for New Energy Materials & Low Carbon Technologies, Tianjin University of Technology, Tianjin 300384, China; 2Key Laboratory of Marine Chemistry Theory and Technology, Ministry of Education College of Chemistry and Chemical Engineering, Ocean University of China, Qingdao 266100, China

**Keywords:** electrocatalysis, carbon dioxide reduction, water oxidation, hetero-binuclear complex, cooperative catalytic effect

## Abstract

One mononuclear Mn(III) complex [Mn^III^**L**(H_2_O)(MeCN)](ClO_4_) (**1**) and one hetero-binuclear complex [(Cu^II^**L**Mn^II^(H_2_O)_3_)(Cu^II^**L**)_2_](ClO_4_)_2_·CH_3_OH (**2**) have been synthesized with the Schiff base ligand (H_2_**L** = N,N′-bis(3-methoxysalicylidene)-1,2-phenylenediamine). Single crystal X-ray structural analysis manifests that the Mn(III) ion in **1** has an octahedral coordination structure, whereas the Mn(II) ion in **2** possesses a trigonal bipyramidal configuration and the Cu(II) ion in **2** is four-coordinated with a square-planar geometry. Electrochimerical catalytic investigation demonstrates that the two complexes can electrochemically catalyze water oxidation and CO_2_ reduction simultaneously. The coordination environments of the Mn(III), Mn(II), and Cu(II) ions in **1** and **2** were provided by the Schiff base ligand (**L**) and labile solvent molecules. The coordinately unsaturated environment of the Cu(II) center in **2** can perfectly facilitate the catalytic performance of **2**. Complexes **1** and **2** display that the over potentials for water oxidation are 728 mV and 216 mV, faradaic efficiencies (FEs) are 88% and 92%, respectively, as well as the turnover frequency (TOF) values for the catalytic reduction of CO_2_ to CO are 0.38 s^−1^ at −1.65 V and 15.97 s^−1^ at −1.60 V, respectively. Complex **2** shows much better catalytic performance for both water oxidation and CO_2_ reduction than that of complex **1**, which could be owing to a structural reason which is attributed to the synergistic catalytic action of the neighboring Mn(III) and Cu(II) active sites in **2**. Complexes **1** and **2** are the first two compounds coordinated with Schiff base ligand for both water oxidation and CO_2_ reduction. The finding in this work can offer significant inspiration for the future development of electrocatalysis in this area.

## 1. Introduction

With the continuous development of humans, the overuse of fossil fuels and the increased emissions of CO_2_ have induced a series of energy shortages and environmental crises [1,2,3,4,5]. To achieve the goal of carbon neutral [6,7,8,9], conversion of CO_2_ into fuels and other chemical sources with high energy density is a very promising strategy [10,11,12,13], which requires the systematic development of advanced electrochemical catalysts. Water oxidation and CO_2_ reduction reactions are two important half-reactions in the classical artificial photosynthesis system, where water oxidation can provide protons and electrons for CO_2_ conversion [14,15,16]. However, both water oxidation and CO_2_ reduction reactions have high kinetic and thermodynamic barriers [17,18,19,20]. Therefore, how to develop cheap, durable, and efficient bifunctional electrochemical catalysts for water oxidation and CO_2_ reduction has become an increasingly important research topic.

Homogeneous catalysis has been widely used in CO_2_ reduction and water oxidation in recent years due to the high activity, facilely adjustable structure and property, and easily clarifying catalytic mechanism [21,22,23,24,25,26,27,28,29,30]. However, so far, only a few complexes have been reported for homogeneously catalyzing both water oxidation and CO_2_ reduction. What has been done in this area to this day is as below. In 2012, Meyer’s team reported a molecular electrochemical catalyst based on Ru(II), which can catalyze CO_2_ reduction and water oxidation simultaneously in a CH_3_CN–H_2_O mixture for the first time [31]. In 2017, Colbran’s group successfully synthesized a Ru(II) complex [(^Bu^tpy)Ru^II^(phenCO_2_)][PF_6_] for both CO_2_ reduction and water oxidation, in which the 1,10-phenanthroline-2-carboxylate ligand was of the greatest significance for the electrocatalysis [32]. In 2021, Wang’s group developed a mononuclear copper complex [Cu^II^**L^1^**_2_](ClO_4_)_2_·2CH_3_OH (**L^1^** = methoxy-di-pyridin-2-ylmethanol) as a bifunctional electrochemical catalyst for CO_2_ reduction and water oxidation, of which the turnover frequencies (TOFs) were 2.99 s^−1^ and 9.20 s^−1^, respectively [33]. Later, they reported a multinuclear cobalt complex (Et_3_NH)_2_[Co^III^_2_Co^II^(OH_2_)(pda)_5_] (H_2_pda = 2,6-pyridinedicarboxylic acid) as an excellent homogeneous electrochemical catalyst for water oxidation and CO_2_ reduction, where the labile aqua ligand played an important role in improving the catalytic performance [34]. Except for the above works, most of the coordination complexes and other materials in this area are heterogeneous catalysts [35,36,37,38,39]. For instance, Lin’s group introduced metal complexes with dicarboxylic acid functional groups into the highly stable and porous Zr^Ⅳ^_6_O_4_(OH)_4_(bpdc)_6_(UiO-67) backbone, achieving excellent performance for water oxidation and CO_2_ reduction [37]. Nam’s group reported the use of N-heterocyclic carbines as persistent connectors for the immobilization of Rubpy complex-based catalyst for photoelectrochemical CO_2_ reduction and water oxidation with high efficiency [38]. In 2016, Morlanés et al. reported the electrolysis of H_2_O/CO_2_ to O_2_/CO with Co^II^FPc as a catalyst immobilized on the carbon electrode, in which the negative induction effect of the fluorine substituent reduces the affinity of the metal center for CO, thus enhancing CO_2_ reduction [39]. In order to simulate the photosynthesis in the natural ecosystem, what is still a huge challenge is how to integrate CO_2_ reduction and water oxidation in a homogeneous catalytic system with high efficiency.

Schiff base ligand features the ability to easily regulate the molecular and electronic structures of complexes, and tune their physicochemical properties [40]. Hence, Schiff base is a type of versatile ligand, which has been widely applied in various fields such as biological [41,42,43,44,45], analytical [46], and catalytic [47,48,49,50,51] fields. Recently, many Schiff base complexes have been utilized in electrocatalysis for CO_2_ reduction or water oxidation. In 2018, Mojtaba et al. reported a novel chromium complex [Cr^III^(**L^2^**-2H)Cl] (**L^2^** is a salen-type Schiff base) with a TOF of 49.7 s^−1^ at an overpotential of 426 mV for water oxidation [52]. In 2022, Mo’s group anchored iron and cobalt ions in the vacancies of the triazine-based Schiff base network, resulting in excellent oxygen evolution reaction (OER) performance with an overpotential of 288 mV [53]. In 2020, Gurpreet and coworkers investigated the activity of mono- and dinuclear complexes of Cr, Mn, Fe, Co, and Ni with macrocyclic Schiff base cuprolic ligand for catalytic CO_2_ reduction, which could efficiently convert CO_2_ to methane at a potential of −0.24 V [54]. In 2021, Bonetto et al. studied the carbon dioxide reduction properties of five Fe(III) complexes of Schiff-base-containing donor sites N_2_O_2_, which show the overpotential of Fe(salen) of 910 mV, *K*_cat_ of 50,000 s^−1^ [55]. Nevertheless, as yet, there has been no reported Schiff base complex acting as bifunctional electrochemical catalyst for CO_2_ reduction and water oxidation. 

Based the reasons above, herein, we have rationally designed and successfully obtained two novel non-noble transition metal coordination complexes: one mononuclear Mn^III^ complex [Mn^III^L(H_2_O)(MeCN)](ClO_4_) (**1**) and one hetero-binuclear Mn^II^–Cu^II^ complex [(Cu^II^LMn^II^(H_2_O)_3_)(Cu^II^L)_2_](ClO_4_)_2_·CH_3_OH (**2**) with the same Schiff base ligand **L**, and explored their homogeneous electrocatalytic properties for water oxidation and CO_2_ reduction. The special coordination environments of **1** and **2** enable both of them to catalyze water oxidation and CO_2_ reduction simultaneously with high catalytic performance. 

Complexes **1** and **2** are the first two compounds coordinated with Schiff base ligand for both water oxidation and CO_2_ reduction. The complexes **1** and **2** show the electrocatalytic activity for water oxidation with TOFs of 3.66 s^−1^ and 7.88 s^−1^ with the overpotentials of 728 mV and 216 mV, meanwhile the catalytic reduction of CO_2_ to CO with the TOF values of 0.38 s^−1^ at −1.65 V and 15.97 s^−1^ at −1.60 V, respectively, which are higher than or comparable with those of reported complexes (Appendix A) [31,32,33,34]. Through great efforts, we used the same Schiff base ligand **L** to obtain mononuclear Mn^III^ complex **1** and hetero-binuclear Mn^II^–Cu^II^ complex **2** for accurately comparing their catalytic behaviors. Complex **2** was found to exhibit superior catalytic performance compared to **1** due to the synergistic catalytic action of the Mn(II) and Cu(II) metal centers in **2**. This research supplies an efficient electrocatalytic system for both water oxidation and CO_2_ reduction with high efficiency, stability, and selectivity under homogeneous conditions, which can offer significant inspiration for the future development of electrocatalysis in this area. 

## 2. Results and Discussion

### 2.1. Characterization of the Crystal Structures

Single-crystal X-ray diffraction analysis demonstrates that **1** and **2** crystallize in the monoclinic system with space group *P*121/*c*1 and triclinic crystal system with *P*-1 space group, respectively (Appendix A). As illustrated in Figure 1, complex **1** is a mononuclear complex, of which the metal center Mn is six-coordinated with two N atoms and two O atoms from the deprotonated Schiff base ligand (**L**), as well as one O atom and one N atom from two labile solvent ligands of water and acetonitrile, respectively, forming a distorted octahedral coordination structure. With a particular configuration, complex **2** consists of three almost planar parts, of which a hetero-binuclear Mn–Cu cation moiety is sandwiched between two identical mononuclear copper natural moieties connected by π–π stacking interaction. In the hetero-binuclear Mn–Cu cation, Cu is four-ligated by two O atoms and two N atoms from the Schiff base ligand **L**, leading to a square-planar coordination geometry, meanwhile, Mn is five-coordinated by two O atoms from **L** and three O atoms from water ligands with a distorted trigonal bipyramid configuration. Additionally, the Mn and Cu centers are linked together by two *µ*_2_-O bridges with a distance of ~3.216 Å. In the two identical mononuclear copper parts, the Cu center is in coplanar configuration chelated with two O atoms and two N atoms from the ligand **L**. According to the bond valence sum calculations, the manganese ion in complex **1** is in a +3 oxidation state, while in complex **2**, the valence states of the three copper ions are all +2 and the manganese ion is +2 [56]. As shown in Appendix A, the magnetic moment (*µ*_eff_) of complex **1** at room temperature is around 5.0 µB, which suggests that the manganese ion should be in the +3 oxidation state. It may be due to the oxidation of the Mn^II^ ion in the air during the reaction process, which is similar to the other reported mononuclear Schiff base manganese complexes (Appendix A) [57]. However, the Mn^II^ ion is not oxidized in the hetero-binuclear Mn–Cu system of complex **2**, which resembles the other reported Schiff base Mn–Cu complexes (Appendix A) [58,59].

### 2.2. Electrochemical Properties under Atmosphere of Ar

Figure 2 shows the cyclic voltammograms (CVs) of complexes **1** and **2** in the DMF solution at the scan rates ranging from 100 to 500 mV s^−1^ under an argon atmosphere (all potentials are relative to the NHE electrode).

While scanning towards the cathode, complexes **1** and **2** exhibit irreversible reduction peaks at *E*_p_ = −1.79 V and −0.87 V, respectively. When scanning towards the anode potential, complex **1** shows three irreversible oxidation peaks at *E*p = 0.21 V, 1.55 V, 1.72 V (Figure 2a), and complex **2** shows four irreversible oxidation peaks at *E*_p_ = −0.76 V, 1.20 V, 1.51 V, 1.71 V (Figure 2b). The peak potentials of both complexes are linearly related to the square root of their scan rates, indicating that the processes are all controlled by diffusion (the insets of Figure 2). 

### 2.3. Electrochemical Properties for Water Oxidation

The electrocatalytic water oxidation performances of complexes **1** and **2** were studied by cyclic voltammetry in DMF:water (3:7 *v*/*v*) solution with 0.2 M phosphate buffer at different pH values. Complex **1** exhibits the largest current density at pH = 7.52, and there are two strongly enhanced irreversible oxidation peaks at 1.27 and 1.58 V (Figure 3a). While complex **2** exhibits the largest current density at pH = 5.39, and there is a strongly enhanced irreversible oxidation peak at 1.15 V (Figure 3b). In addition, during electrolysis at these potentials, many bubbles can be observed on the surface of the GC electrode. The generation of oxygen can be detected by the Ocean Optics NeoFox-GT oxygen sensor. These experimental results indicate that complexes **1** and **2** can catalyze the oxidation of water to produce O_2_ at these potentials. Furthermore, the water oxidation onset potentials of the two complexes are linearly dependent on pH values (the insets of Figure 3) with the slopes of 47 mV pH^−1^ and 64 mV pH^−1^ for complex **1** as well as −55 mV pH^−1^ for complex **2**, respectively, which indicate that the catalytic water oxidation reactions should be all a PCET (proton-coupled electron transfer) process, which is characteristic for a 1e^−^/1H^+^-coupled redox process according to the Nernst equation [60]. According to the Equations (1) and (2), the overpotentials for electrocatalytic water oxidation of complex **1** are 411 and 728 mV, and for complex **2** is 216 mV, which is lower than many previously reported great coordination catalysts [61,62,63,64,65].
𝐸_𝑡ℎ𝑒𝑜𝑟𝑦_ (𝑂_2_/𝐻_2_𝑂) = 1.23 − 0.059 𝑝𝐻(1)
𝜂 = 𝐸_𝑟𝑒𝑎𝑙_ (𝑂_2_/𝐻_2_𝑂) − 𝐸_𝑡ℎ𝑒𝑜𝑟𝑦_ (𝑂_2_/𝐻_2_𝑂)(2)

In order to further evaluate the electrocatalytic reactivity of complexes **1** and **2** for water oxidation, the controlled potential electrolysis (CPE) experiments were carried out in DMF:water (3:7 *v*/*v*) solution with 0.2 M phosphate as buffer at the potentials of 1.27 and 1.58 V for complex **1** at pH = 7.52, and at the potential of 1.15 V for complex **2** at pH = 5.39, respectively, using F-doped tin oxide (FTO) as the working electrode. In the meantime, to determine the generated oxygen, the oxygen sensor was used to record the oxygen density in the solution in situ (the oxygen density in the headspace of the electrolytic cell is below the detection limit of the sensor). It is found that the oxygen content and detected current density in the blank solution are close to zero without complexes (Figure 4, black lines), indicating that no catalysis occurred with no complex **1** or **2** dissolved in the solution. With **1** or **2** dissolved in the system, the current density and oxygen content increase sharply, indicating that the two complexes are excellent electrochemical catalysts for water oxidation at some particular pH values. Throughout an electrolysis of 3600 s at the applied potentials of 1.27 V and 1.58 V for complex **1**, the observed dissolved oxygen concentrations are 238 and 320 μM (Figure 4b), while the current densities are 0.31 and 0.38 mA cm^−2^, respectively (Figure 4a). Based on the experimental results, the Faraday efficiencies (FEs) are calculated to be 84% and 88%, respectively. What is more, throughout an electrolysis of 3600 s at the applied potential of 1.15 V for complex **2**, the observed dissolved oxygen concentration is 157 μM (Figure 4d), while the current density is 0.22 mA cm^−2^ (Figure 4c), and the FE is 92%. Moreover, the current densities of both complexes tend to be stable during electrolysis, indicating that they remain stable during the electrolysis process. Additionally, the rinse tests can also verify the high stability of the two catalysts (Figure 4). In addition, to further explore the stability of the complexes during electrocatalytic water oxidation (Appendix A), in situ UV–Vis spectroelectrochemical detections were performed for the two complexes. The spectrums detected every 500 s are almost the same for both **1** and **2**, confirming that the two complexes remain stable during the catalytic processes.

In addition to the above studies, the relationship between the concentrations of complexes **1** and **2** and the peak currents is also explored for kinetic studies. The peak currents raise continuously with the increase of the concentration of the complexes, which shows linear relationships, indicating that the rate-determining steps of the water oxidation processes for the two complexes are all in the first order (Appendix A). To further investigate the kinetic information of water oxidation catalyzed by complexes **1** and **2**, the CV curves of the two complexes at different scan rates were determined at the optimal pH (Figure 5). The reaction rate coefficient *k*_cat_ was calculated based on Equation (3), which can be roughly considered as the turnover frequency (TOF) of the catalytic reaction. As shown in the insets of Figure 5, TOFs are calculated to be 1.92 s^−1^ and 3.66 s^−1^ for complex **1** at the potentials of 1.27 V and 1.58 V, respectively, and 7.88 s^−1^ for complex **2** at 1.15 V, which is higher than many other great catalysts (Appendix A). Complex **2** shows much greater catalytic performance for water oxidation than that of complex **1**, which could be due to the unique coordination sphere of **2** and the cooperative catalytic effect between the manganese and copper ions in complex **2**.
(3)$$\frac{i_{cat}}{i_{p}}=2.242n_{cat}{\left(\frac{k_{cat}RT}{F}\right)}^{\frac{1}{2}}v^{-\frac{1}{2}}$$

In this equation, *i*_cat_ is the catalytic current, *i*_p_ is the peak current measured without substrate, *n*_cat_ is the number of electrons involved in the catalytic reaction, *F* is the Faraday constant, *R* is the general gas constant, *T* is the Kelvin unit temperature, *v* is the scan rate.

### 2.4. Electrochemical Properties for Carbon Dioxide Reduction

To investigate the performance of the electrocatalytic CO_2_ reduction of complexes **1** and **2**, cyclic voltammetry and controlled potential electrolysis (CPE) experiments were performed using the glassy carbon (GC) and FTO electrode, respectively, in a DMF solution containing 0.1 M ^n^Bu_4_NPF_6_ under a saturated CO_2_ atmosphere. Cyclic voltammetric curves under an Ar and CO_2_ atmosphere at the same scan rate show significant enhancement at the reduction peaks around −1.65 V for complex **1** and −1.60 V for complex **2** (Figure 6), indicating that both complexes can catalyze CO_2_ reduction. Furthermore, by testing the CVs of the two complexes at different scan rates, it is found that the peak currents are linearly related to the square root of the scan rates (Appendix A), indicating that the catalytic CO_2_ reduction processes by complexes **1** and **2** are both diffusion controlled. In addition, the CV plots at different scan rates are well reproducible and no new reduction peaks appeared, which further affirms the high stability of the two complexes during catalysis. Additionally, the relationship between the concentrations of the complexes and the peak potentials was investigated by CVs (Appendix A). It is found that the concentrations of **1** and **2** exhibit a linear relationship with the peak currents, revealing that the rate-determining steps for the two catalytic reactions are all in the first order.

The effect of proton source on the electrocatalytic CO_2_ reduction of the complexes was further investigated by adding distilled water to 0.1 M ^n^Bu_4_NPF_6_ DMF, which is shown in Figure 7. The current densities for complexes **1** and **2** raise with the increasing amount of water added, indicating that the proton source could improve the catalytic activity of both complexes. The TOFs of **1** and **2** for electrocatalytic CO_2_ reduction are obtained based on Equation (4), which are 0.38 s^−1^ for complex **1** at the potential of −1.65 V, and 15.97 s^−1^ for complex **2** at the potential of −1.60 V, which is higher than many other reported catalysts (Appendix A).
(4)$$TOF=\left(\frac{Fvn_{p}^{3}}{RT}\right){\left(\frac{0.4463}{n_{cat}}\right)}^{2}{\left(\frac{i_{cat}}{i_{p}}\right)}^{2}$$

In this formula, *F* is the Faraday constant (96,485 C mol^−1^), *v* is the scan rate (0.1 V s^−1^), *n*_p_ is the number of electrons participating in the noncatalytic redox reaction (*n*_p_ = 1), *R* is the gas constant (8.314 J K^−1^ mol^−1^), *T* is the temperature (298.15 K), *n*_cat_ is the number of electrons involved in the catalytic reaction (*n*_cat_ = 2, reducing CO_2_ to CO), and *i*_p_ and *i*_cat_ are the peak currents under Ar and CO_2_, respectively.

To further investigate the reactivity of **1** and **2** for electrocatalytic CO_2_ reduction, controlled potential electrolysis experiments (CPEs) were performed in 0.1 M ^n^Bu_4_NPF_6_ DMF solution under a CO_2_-saturated atmosphere using FTO as the working electrode, and the electrolysis products were analyzed by gas chromatography. The current density of complex **1** can reach −0.069 mA cm^−2^ during the electrolysis of 3600 s at the applied potential of −1.65 V. After the introduction of water as a proton source, the current density can reach −0.10 mA cm^−2^ (Figure 8a). At the applied potential of −1.60 V, the current density of complex **2** can reach −0.31 mA cm^−2^ during 3600 s of electrolysis and −0.45 mA cm^−2^ after the addition of water (Figure 8b). Based on gas chromatography analysis, it is revealed that complexes **1** and **2** can electrocatalyze CO_2_ reduction to CO. The FEs are 2% and 36% for complexes **1** and **2**, respectively, and up to 4% and 48% after the addition of 4.5 M water (Figure 9). Furthermore, to further investigate the stability of the complexes during electrocatalytic carbon dioxide reduction (Appendix A), in situ UV–Vis spectroelectrochemistry was carried out for the two complexes. The spectra are almost the same during 3600 s catalysis, confirming that **1** and **2** stay stable during the catalytic process. Notably, complex **2** displays better catalytic behavior for CO_2_ reduction than that of complex **1**, which could be because of the special configuration of complex **2** determined by the Schiff base ligand together with the synergistic catalytic effect between the adjacent Mn and Cu active sites in the complex.

## 3. Materials and Methods

### 3.1. Materials and Characterization

o-Vanillin (C_8_H_8_O_3_, AR) was purchased from Aladdin Biochemical Technology Co., Ltd., (Shanghai, China). 1,2-diaminobenzene (C_6_H_8_N_2_, AR) was purchased from Shanghai Macklin Biochemical Co., Ltd. Manganese(II) Perchlorate (Mn(ClO_4_)_2_·6H_2_O, AR) was purchased from Energy Chemical Reagent Co., Ltd., (Shanghai, China). Cupric Acetate Monohydrate (Cu(CH_3_COO)_2_·H_2_O, AR), Acetic Acid (CH_3_COOH, AR), methanol (CH_3_OH, AR), and N,N-Dimethylformamide (C_3_H_7_NO, AR) were purchased from Sinopharm Chemical Reagent Co., Ltd., (Shanghai, China). Deionized water was utilized in all experiments. Unless specified otherwise, all chemicals were used as received without purification. 

Elemental analyses (C, H, and N) of complexes were performed on a model 2400 PerkinElmer analyzer. The infrared spectra (2 wt% sample in KBr pellets) were recorded on a Nicolet 170SX spectrometer in the 4000–400 cm^−1^ region. UV–Vis absorption spectra were recorded on TU-1800. Single crystal was mounted on a Bruker SMART APEX II CCD X-ray single-crystal diffractometer, and all data were collected at 173 K with graphite monochromated MoKa radiation (λ = 0.71073 Å) in I > 2σ(I) diffraction spots and reduced by the SAINT program, and absorption corrections were applied using the program SADABS [66]. Structures for **1** and **2** were solved by direct methods using the SHELXS-2014 package and refined with SHELXL-2014/7 [67]. CCDC numbers for the complexes **1** and **2** are 2069590 and 2069498, respectively.

### 3.2. Synthesis

#### 3.2.1. Synthesis of N,N′-bis(3-Methoxysalicylidene)-1,2-Phenylenediamine (H_2_L)

The pro-ligand H_2_**L** N,N′-(1,2-phenylene)-bis(3-methoxysalicylaldimine) was synthesized according to the literature procedure [68]. 

The synthesis scheme for H_2_**L** and the metal complexes **1** and **2** was illustrated in Appendix A.

#### 3.2.2. Synthesis of [Mn^III^L(H_2_O)(MeCN)](ClO_4_) (1)

In a 50 mL flask, the ligand H_2_**L** (0.0376 g, 0.1 mmol) was dissolved in 5 mL of methanol, which was then added dropwise to Mn(ClO_4_)_2_·6H_2_O (0.0362 g, 0.1 mmol) in 10 mL of acetonitrile solution. Afterwards, the reaction mixture was stirred for 2 h at 25 °C, and then filtered. X-ray quality crystals were grown by vapor diffusion of diethyl ether into the resulted filtrate at 25 °C. The large quantity reddish-brown crystals were obtained after one week, which were washed with diethyl ether and dried in vacuum at 25 °C, affording 0.0260 g of complex **1** (72% yield, based on Mn). Elemental analysis of Calc. (Found) for C_24_H_23_ClMnN_3_O_9_: C, 48.78 (48.99); H, 3.74 (3.91); N, 7.53 (7.14). IR (KBr disk, cm^−1^): 619, 1086, 1257, 1434, 1598, 1648, 2343 (Appendix A). 

#### 3.2.3. Synthesis of [(Cu^II^LMn^II^(H_2_O)_3_)(Cu^II^L)_2_](ClO_4_)_2_·CH_3_OH (2)

In a 50 mL conical flask, Cu(CH_3_COO)_2_·H_2_O (0.0200 g, 0.1 mmol) was first dissolved in 1 mL H_2_O, which was added dropwise to the ligand H_2_**L** (0.0376 g, 0.1 mmol) dissolved in 10 mL of methanol solution. Afterwards, the reaction mixture was stirred for 2 h at 25 °C. Then, Mn(ClO_4_)_2_·6H_2_O (0.0362 g, 0.1 mmol) dissolved in a solution of methanol (5 mL) was added dropwise to the above resulted mixture, which was stirred for 2 h at 25 °C, and then filtered. X-ray quality crystals were grown by vapor diffusion of diethyl ether into the resulted filtrate at 25 °C. The large quantity brownish-yellow crystals were obtained after one week, which were washed with diethyl ether and dried in vacuum at 25 °C, affording 0.0221 g of complex **2** (61% yield, based on Mn). Elemental analysis of Calc. (Found) for C_67_H_64_Cl_2_Cu_3_MnN_6_O_24_: C, 48.33 (48.62); H, 3.94 (3.87); N, 5.43 (5.08). IR (KBr disk, cm^−1^): 410, 524, 669, 1093, 1257, 1446, 1604, 1680, 2461 (Appendix A).

#### 3.2.4. Electrochemical Measurement and Electrolytic Product Analysis

All electrochemical experiments were tested by a CHI660E electrochemical analyzer to study their electrocatalytic properties and performed in a single chamber three-electrode cell. The electrocatalytic water oxidation properties of the complexes were measured in DMF:water (3:7 *v*/*v*) solution with 0.2 M phosphate buffer, and the electrocatalytic CO_2_ reduction properties were measured in 0.1 M ^n^Bu_4_NPF_6_ DMF solution with distilled water as proton sources. The solution was purged with high purity argon gas for 30 min prior to each electrochemical test to eliminate air interference.

For the cyclic voltammetry (CV) experiments, the glassy carbon electrode, Ag/AgCl electrode, and platinum wire electrode were used as the working electrode, reference electrode, and counter electrode, respectively. The glassy carbon electrode was polished several times with polishing powder, then washed with anhydrous ethanol and deionized water, and finally dried before use.

For the controlled potential electrolysis experiments (CPE), the working electrodes were replaced with F-doped tin oxide (FTO) conductive glass substrates (1 × 1 cm, effective surface area of 1.0 cm^2^), which were immersed in ethanol solution containing 5% NaOH for several hours before use, and then washed with water, ethanol, and water in turn. For water oxidation, the pH was adjusted using 0.1 M sodium phosphate and 0.1 M phosphoric acid. For CO_2_ reduction, distilled water was added to 0.1 M ^n^Bu_4_NPF_6_ DMF solution to tune the concentration of proton sources.

The oxygen concentration generated during the water oxidation electrolysis process was detected using the Ocean Optics NeoFox-GT oxygen sensor, and the carbon monoxide concentration generated during CO_2_ reduction was detected using the Shimadzu GC-2014. 

## 4. Conclusions

In this work, we have successfully isolated two novel non-noble transition metal complexes with the same Schiff base ligand **L**: mononuclear Mn^III^ complex [Mn^III^**L**(H_2_O)(MeCN)](ClO_4_) (**1**) and hetero-binuclear Mn^II^–Cu^II^ complex [(Cu^II^**L**Mn^II^(H_2_O)_3_)(Cu^II^**L**)_2_](ClO_4_)_2_·CH_3_OH (**2**). The structures of the two complexes are studied by X-ray crystallography, which reveals that the Mn^III^ center in **1** is six-coordinated, while in complex **2**, the sphere of Cu^II^ center is in the square-planar coordination sphere and the Mn^II^ center is in a five coordination environment. The peculiar coordination environments of **1** and **2** make them capable of homogeneous electrocatalytic water oxidation and CO_2_ reduction simultaneously. Complexes **1** and **2** show water oxidation reactivity with the overpotentials of 728 mV and 216 mV, FEs of 88% and 92%, respectively, while the TOF values for the catalytic reduction of CO_2_ to CO are 0.38 s^−1^ at −1.65 V and 15.97 s^−1^ at −1.60 V, respectively. Complexes **1** and **2** are the first two compounds coordinated with Schiff base ligand as bifunctional electrocatalysts in this field. By comparison, complex **2** displays much greater catalytic performance than that of **1** and many other reported complexes. The excellent catalytic activity of complex **2** could be due to the special configuration of complex **2** determined by Schiff base ligand together with the synergistic catalytic effect between the adjacent Mn and Cu active sites in the complex. This research supplies a facile strategy of the synergistic catalytic effect between two different centers for investigating efficient non-noble transition metal complexes as bifunctional electrochemical catalysts for water oxidation and CO_2_ reduction.

## Figures and Tables

**Figure 1 molecules-28-01074-f001:**
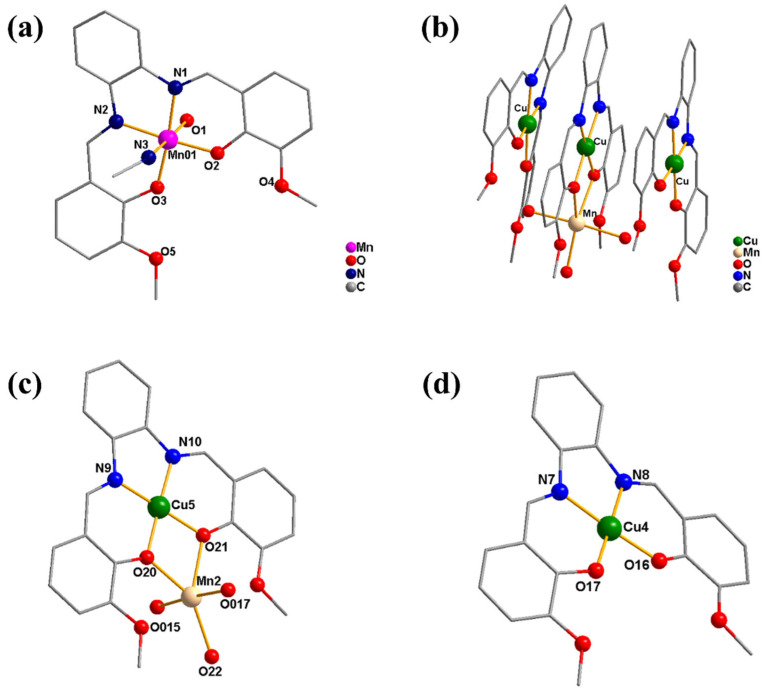
Crystal structures of complex **1** (**a**) and **2** ((**b**): the whole part of **2**; (**c**): the middle Mn–Cu cation moiety in **2**; (**d**): the natural Cu moiety in **2**) (hydrogen atoms, solvents, and anions have been omitted for clarity).

**Figure 2 molecules-28-01074-f002:**
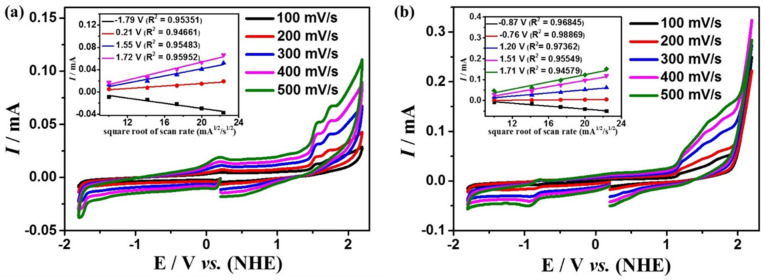
Cyclic voltammogram of 0.5 mM complexes **1** (**a**) and **2** (**b**) in the DMF solution, the sweep speed range is 100–500 mV s^−1^. The insets of (**a**,**b**) are the linear relationship between the irreversible peak current and the square root of the scan rate mA^1/2^/s^1/2^ at different potentials for **1** and **2**, respectively.

**Figure 3 molecules-28-01074-f003:**
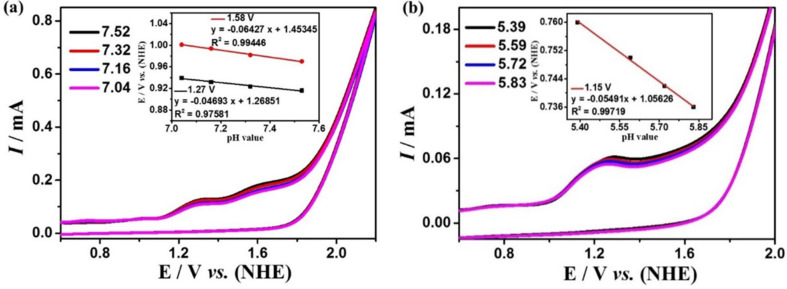
Cyclic voltammograms of 0.5 mM complexes **1** (**a**) and **2** (**b**) in DMF:water (3:7 *v*/*v*) solution with 0.2 M phosphate buffer at different pH (scan rate 100 mV s^−1^). The insets of (**a**) and (**b**) are the relationships of onset potentials for electrocatalytic water oxidation with pH values.

**Figure 4 molecules-28-01074-f004:**
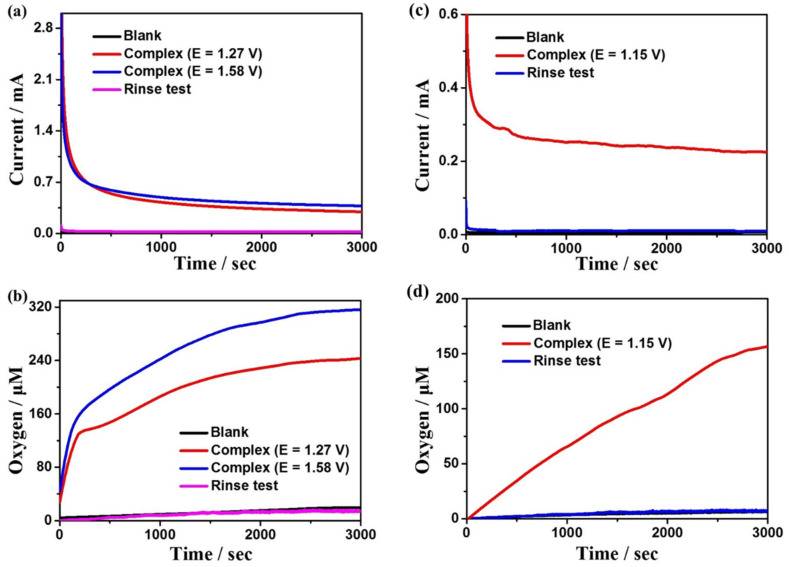
(**a**) CPEs for a solution containing 0.5 mM complex **1** and no complex **1** on the FTO working electrode (1.0 cm^−2^). (**b**) Dissolved oxygen concentration curves of complex **1** during electrolysis. (**c**) CPEs for a solution containing 0.5 mM complex **2** and no complex **2** on the FTO working electrode (1.0 cm^−2^). (**d**) Dissolved oxygen concentration curves of complex **2** during electrolysis.

**Figure 5 molecules-28-01074-f005:**
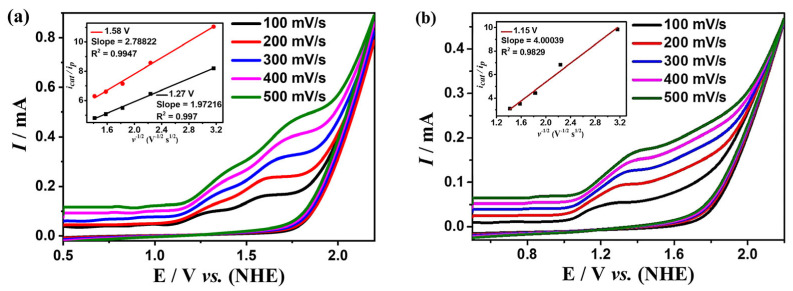
Cyclic voltammogram of 0.5 mM complexes **1** (**a**) and **2** (**b**) in DMF:H_2_O (3:7 *v*/*v*) solution containing 0.2 M phosphonate buffer, the sweep speed range is 100–500 mV s^−1^. The inset is the relationships of the values of *i*_cat_/*i*_p_ with the inverse function of square root of sweep velocity.

**Figure 6 molecules-28-01074-f006:**
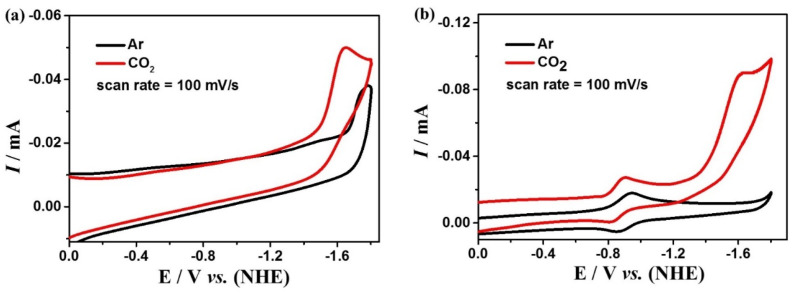
Cyclic voltammetry of 0.5 mM complexes **1** (**a**) and **2** (**b**) in the DMF solution under CO_2_ (red) and Ar (black) at glassy carbon.

**Figure 7 molecules-28-01074-f007:**
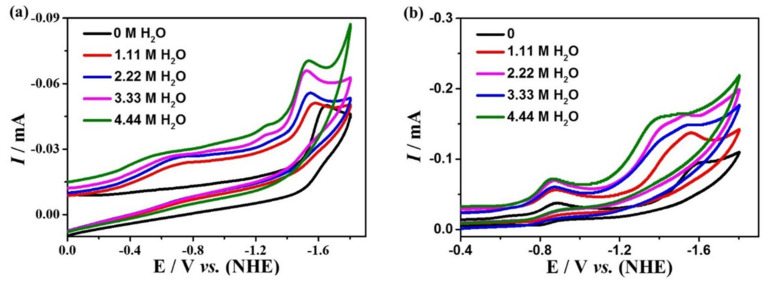
Cyclic voltammetry of 0.5 mM complexes **1** (**a**) and **2** (**b**) in different concentrations of H_2_O in 0.1 M ^n^Bu_4_NPF_6_ DMF solution.

**Figure 8 molecules-28-01074-f008:**
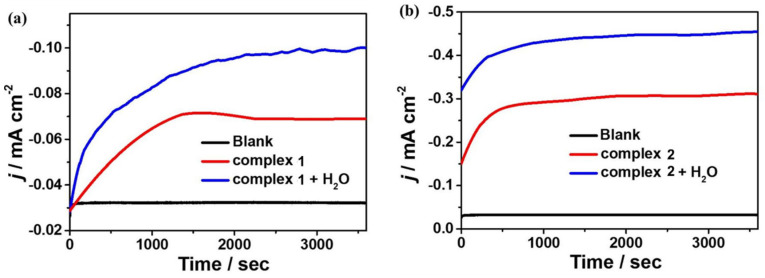
CPE curves of 0.5 mM complexes **1** (**a**) and **2** (**b**) in 0.1 M ^n^Bu_4_NPF_6_ DMF solution with and without H_2_O and 0.1 M ^n^Bu_4_NPF_6_ DMF solution without complex **1** or **2**.

**Figure 9 molecules-28-01074-f009:**
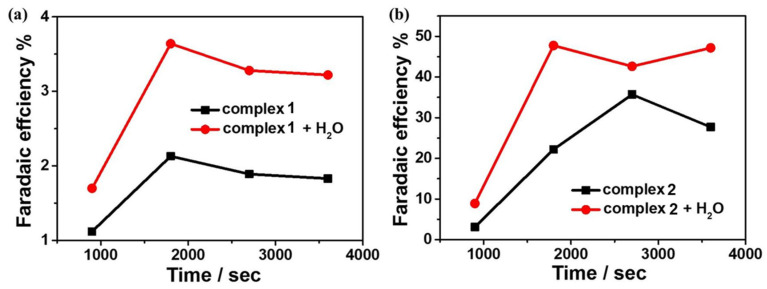
(**a**) Faraday efficiency of carbon monoxide produced by 0.5 mM complex **1** in the presence/absence of H_2_O in DMF solution; (**b**) Faraday efficiency of carbon monoxide produced by 0.5 mM complex **2** in the presence/absence of H_2_O in DMF solution.

## Data Availability

Data available on request to the corresponding author.

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
