# Peer review of "Two Novel Schiff Base Manganese Complexes as Bifunctional Electrocatalysts for CO2 Reduction and Water Oxidation"

_molecules, 2023, doi:10.3390/molecules28031074_

Round 1

Reviewer 1 Report

This manuscript wants to present a bifunctional electrocatalyst for CO2 reduction and water oxidation. It was found that two non-noble transition metal coordination complexes: one mononuclear MnIII complex [MnIIIL(H2O)(MeCN)](ClO4) (1) and one heterometallic MnII-CuII complex [(CuIILMnII(H2O)3)(CuIIL)2](ClO4)2·CH3OH (2) with the same Schiff base ligand L, were constructed, especially, complex 2 shows better electrocatalytic performance in comparison with complex 1. I support the publication of this work in molecules after the followed revisions.

1. The title should be revised, because OER & CO2RR were separately studied in this work.

2. Introduction section should be improved, especially, the novelty and significancy should be clearly emphasized in the introduction section.

3. How about the performance comparison (OER & CO2RR) between as-obtained catalysts and the reported similar homogeneous catalysts? It is very important to demonstrate the advance of as-obtained catalysts in comparison with the others in literatures.

4. How about the crystal structures of complex 1 and 2 after the electrocatalysis?

5. Some very related literatures may be incited to enrich the content of this work, such as, Journal of Electroanalytical Chemistry 928 (2023) 117089; Chinese Chemical Letters 34 (2023) 107119.

6. English writing should be improved.

Reviewer 2 Report

The article is very interesting and promising for science, since they present a heterometallic complex of Mn-Cu [(CuIILMnII(H2O)3)(CuIIL)2](ClO4)2 CH3OH (2) have been synthesized with the base ligand Schiff's (H2L=N,N'-bis(3-methoxysalicylidene)-1,2henylenediamine). They show that the two complexes can electrocatalyze the oxidation of water and the CO2 reduction simultaneously. Complexes 1 and 2 show that the overpotentials for the oxidation of water are 728 mV and 217 mV, the FE are 88 % and 92 %, respectively, as well as the TOF values for the catalytic reduction of CO2 to CO are 0.38 s. -1 at -1.65 V and 15.97 s-1 at -1.60 V, respectively. However, the article presents some observations that can improve this work.

1.- It is necessary to add more references in the introduction.

2.- It is necessary to add a paragraph in the introduction that I planted which is the contribution of this article.

3.- It is necessary to add a paragraph at the end of the introduction about the section of this work.

4.-The director should NOT enter the results and discussion, a section on materials and methods used for certain experiments should be added.

5.- A nomenclature table must be added

6.-The materials and methods section should go before results and discussion.

7.- The wording of how the experimentation was done should be written in a more technical way and not in a colloquial way.

The article can be accepted after having attended the observations presented.

Reviewer 3 Report

After careful review of this manuscript, I would like to clarify the following:

This work describes the synthesis of two manganese complexes with Schiff bases, one mononuclear containing only manganese(III) ions and the other dinuclear containing manganese(II) and copper(II) ions. The authors relied on proving the structures of the current metal complexes only on measurements X-ray structural analysis of suitable single crystal.

The metal complexes were employed in catalytic processes for the hydrolysis of water and the reduction of CO2 to CO by electrochemical technique. The topic of the work is useful and interesting and is of interest to many who are interested in coordination and electrochemical chemistry, catalysis, and those interested in environmental studies.

Given the importance of this work, I recommend that this work be presented to a specialist in the field of electrochemical chemistry and electrocatalysis so that he can provide an accurate and specialized review of the electrocatalysis part, which occupies the greatest share of this manuscript.

The English language of the manuscript is in need of careful revision, and I have tried as much as possible to correct it.

Attached are some reviews that can be used:

1) In the Abstract part, I made some modifications in blue, as follows:

Abstract: One mononuclear Mn(III) complex [MnL(H2O)(MeCN)](ClO4) (1) and one hetero-binuclear complex [(CuIILMnII(H2O)3)(CuL)2](ClO4)2·CH3OH (2) have been synthesized with the Schiff base ligand (H2L = N,N'-bis(3-methoxysalicylidene)-1,2-phenylenediamine). Single crystal x-ray structural analysis manifests that Mn(III) ion in 1 has an octahedral coordination structure, whereas, Mn(II) ion in 2 possesses a trigonal bipyramidal configuration and Cu(II) ion in 2 are four-coordinated with a square-planar geometry. The coordination environment of Mn(III), Mn(II) and Cu(II) ions in 1 and 2 provided by the Schiff base ligand (L) and labile solvent molecules. The coordinately unsaturated environment of Cu(II) center in 2 can perfectly facilitate the catalytic performance of 2. Electrochimerical catalytic investigation demonstrates that the two complexes can electrochemically catalyze water oxidation and CO2 reduction simultaneously. Complexes 1 and  display that the over potentials for water oxidation are 728 mV and 217 mV, FEs are 88% and 92%, respectively, as well as the TOF values for catalytic reduction of CO2 to CO are 0.38 s-1 at -1.65 V and 15.97 s-1 at -1.60 V, respectively. Complex 2 shows much better catalytic performance for both water oxidation and CO2 reduction than that of complex 1, which could be owing to structural reason which is attributed to the synergistic catalytic action of neighboring Mn(III) and Cu(II) active sites in 2.

2- This is a list of some phrases and words that should be changed in the introduction part:

Society should changes to human

Electrocatalysts should changes to electrochemical catalysts

Physic-chemical properties should changes physicochemical

Electrocatalyst should change to electrochemical catalyst through out the manuscript

Schiff base Fe(LN2O2) should change to Fe(III) complexes of Schiff base containing donor sites N2O2.

Heterometallic should change to hetero binuclear through out the manuscript

Through thoroughly comparing their electrocatalytic behaviors, it is found that complex 2 shows superior catalytic performance to 1 due to the peculiar configuration of 2 and the synergistic catalytic effect of the metal centers Mn and Cu in 2.

The above sentence should change to:

By accurately comparing their electrochemical catalytic behaviors, complex 2 was found to exhibit superior catalytic performance compared to 1 due to the synergistic catalytic action of the Mn(II) and Cu(II) metal centers in 2.

3) Cr, Mn, Fe, Co and Ni; the symbols for these elements are mentioned in the introduction, and their oxidation state must be clarified by writing a Roman numeral in brackets without a space, and please take into account this through out the manuscript.

4) FEs, TOF, OER, … and many abbreviations are included through out the manuscript; please write the original term, followed by the abbreviation in brackets, when it is mentioned the first time.

In order for the current reviewer to be convinced of the structures of the metal complexes under study, the authors are requested to include the following:

1- Scheme for the synthesis of Schiff bases under the study - taking into account the detailed description of the synthesis of the current Schiff bases in the experimental section.

2- Preparing drawings using the Chem Draw program, with schemes for synthesis of Schiff bases, as well as for the metal complexes under study, to be included in appropriate locations in the Results and Discussions section.

3- Please provide a better drawing than Figure 1 describing the structure of the current metal complexes, because the current figure 1 is not suitable.

4) The important question is how did the authors prove that the manganese(II) was oxidized to the manganese(III) in the case of complex 1, given that manganese(II) salt Mn(ClO4)2 was used. There is no experimental measurement or evidence that confirms or denies this, such as measurements of magnetic susceptibility. In the same regard, in the event that the manganese(III) ion in complex 1 is trivalent and the coordinated ligand is dibasic, where is the anion equivalent to the third charge of this manganese(III) ion - it must be reviewed and clarified. In the same vein, why was the manganese(II) ion not oxidized in the case of complex 2?

In light of the aforementioned, the current reviewer cannot recommend publishing this manuscript in its current state, which needs substantial revisions.

Reviewer 4 Report

Referee’s Report

Manuscript Title: Electrocatalytic CO2 reduction coupled with water oxidation by

two Schiff base manganese complexes

Recommendation: Major Revision

Comments to Authors:

The work is related to the journals aims and scope. There are some Major issues with the manuscript and some of them are indicated below. The manuscript can be accepted, if the authors revise their manuscript in accordance to the following comments:

1. Revise the title such that it would be objective, and capable to showcase the research question. Divide the title into major three or four keywords. Each keyword should be introduced and formed the basis of each research question. Enrich the novelty and objective portion of your study

2.  Research questions are needed. This would guide the author to structure logical analysis of results. Logical questions are expected. This would help readers to link what is known in the literature with the novelty of this study.

4. The originality of the paper needs to be stated clearly. It is of importance to have sufficient results to justify the novelty of a high-quality journal paper. The Introduction should make a compelling case for why the study is useful along with a clear statement of its novelty or originality by providing relevant information and providing answers to basic questions such as:

a)     What is already known in the open literature?

b)    What is missing (i.e., research gaps)?

c)     What needs to be done, why and how?

d)    Clear statements of the novelty of the work should also appear briefly in the Abstract and Conclusions sections.

6. The conclusion must answer whether the proposed method can solve the research problem and achieve the objective.

7. Authors have ignored some of the recent publications of the related work. Authors should remove the old references and update them with new one. Discuss these articles in the introduction section:

a)  Cattaneo–Christov heat flux model in Darcy–Forchheimer radiative flow of MoS2–SiO2/kerosene oil between two parallel rotating disks. https://doi.org/10.1007/s10973-022-11248-0

b)  Analysis of Heat Transfer of Mono and Hybrid Nanofluid Flow between Two Parallel Plates in a Darcy Porous Medium with Thermal Radiation and Heat Generation/Absorption. Symmetry, 14(9), 1943 https://doi.org/10.3390/sym14091943

c)  Heat transfer in micropolar hybrid nanofluid flow past a vertical plate in the presence of thermal radiation and suction/injection effect, https://doi.org/10.1016/j.padiff.2021.100240

Round 2

Reviewer 3 Report

Dear Editor Journal molecules

Thank you for your kind invitation to review the revised version of manuscript ID: molecules-2166486 and entitled: Two Novel Schiff Base Manganese Complexes as Difunctional Electro Catalysts for CO2 Reduction and Water Oxidation. I have reviewed the revised version of the manuscript and the corrections provided are satisfactory and I therefore recommend that the manuscript be accepted in its current state for publication.

Reviewer 4 Report

Paper has been accepted in the present form.